# Relation between Weight Status, Physical activity, Maturation, and Functional Movement in Adolescence: An Overview

**DOI:** 10.3390/jfmk4020031

**Published:** 2019-05-30

**Authors:** Josip Karuc, Marjeta Mišigoj-Duraković

**Affiliations:** Faculty of Kinesiology, University of Zagreb, Horvaćanski zavoj 15, 10 000 Zagreb, Croatia

**Keywords:** FMS, pubescence, pediatric population, fundamental movement

## Abstract

Obesity, low level of physical activity and dysfunctional movement patterns presents one of the leading health issues that can contribute to increased risk for developing not only metabolic and cardiovascular disease, but also musculoskeletal problems. The aim of this paper is to summarize literature and evidence about relationship between functional movement (FM) patterns, physical activity (PA) level and weight status in average adolescent population. In addition, this paper summarized current evidence about relations between maturation effects and functional movement among athletic adolescent populations. Summary of current evidence suggests that decreased physical activity level is negatively correlated to functional movement in adolescence. Additionally, most studies suggest that weight status is negatively correlated to functional movement patterns although there is conflicting evidence in this area. Evidence consistently showed that overweight and obese adolescents exhibit poorer functional movement compared to normal weight adolescents. In addition, it appears that maturation has effects on functional movement in athletic populations of adolescents. It is therefore important that practitioners consider interventions which develop optimal functional movement alongside physical activity and weight management strategies in children, in order to reduce the risks of injuries and pathological abnormality arising from suboptimal movement patterns in later life.

## 1. Introduction

According to WHO, obesity has tripled since 1975 and thus represents one of the leading world health problems [1]. Along with the obesity and overweight, low level of physical activity (PA) puts overweight children at a higher risk for developing noncommunicable diseases. Although PA level is important for health of the locomotor system and represents a quantitative measure of human movement, due to importance of locomotor health, qualitative aspects of movement need to be considered as well. Functional movement (FM) considers qualitative aspects of movement, and can be defined as optimal postural control and mobility of joints and body regions involved in a particular movement. On the other hand, dysfunctional movement patterns present low level of quality of FM and can be related to injury incidence and thus endanger musculoskeletal (MSK) health and can contribute to developing degenerative changes in adulthood.

Looking altogether, obesity, low level of PA, and dysfunctional movement, can contribute to even more increased risk for developing not only metabolic and cardiovascular disease, but also MSK problems in adulthood. Therefore, literature about mutual adverse effects of dysfunctional movement on mentioned variables need to be considered in order to provide practical information for professionals in the field of kinesiology, medicine, and related areas. 

The importance and influence of PA and weight status on adolescent health has been investigated widely, however only few studies have examined the relationship between FM, PA level, and obesity among the pediatric population. To date, only few studies have examined FM in the general adolescent population [2,3,4,5,6,7,8,9,10,11]. There are only four studies that investigated relation between weight status and FM in children [8,9,10,11]. Additionally, only one study by Duncan and Stanley, investigated the association between FM and PA level among children [10]. However, no study appears to have summarized literature about functional movement and its relation to PA level, weight status, and maturation in the average adolescent population.

Therefore, the aim of this paper is to summarize literature and evidence about the relationship between FM patterns, PA level, and weight status in the average adolescent population. Additionally, this paper will summarize current evidence about relations between maturation effects and functional movement among the athletic adolescent population.

## 2. Materials and Methods

The author of this study conducted a search in PUBMED (from 1 January 1990 to 1 May 2019) searching for association between functional movement, weight status, physical activity level, and maturation in adolescent populations. Key words used for electronic searches were: “functional movement”, “functional movement screen”, “weight status”, “physical activity”, “physical activity level”, and “maturation”. Combining the key words “functional movement” and “functional movement screen” with the other key words: “adolescents”, “intervention”, “weight status”, “physical activity”, “physical activity level”, and “maturation” were used according to Boolean logic. The studies were checked by one researcher (J.K.) and were searched by title/abstract. In this study, inclusion criteria were (1) studies that investigated functional movement on average adolescent population and exercise intervention aimed to improve functional movement outcomes in average adolescents, (2) studies that examined the association between functional movement and weight status and physical activity level in average adolescent populations, (3) studies that investigated the association between functional movement and maturation in athletic adolescent populations, and (4) English as a publication language. Exclusion criteria in this study were (1) studies that investigated adult populations, special populations (e.g., firefighters, officers, military population etc.) and populations with specific diseases or injuries, and (2) studies that investigated functional movement skills (FMS) or functional capacity/competence in pediatric populations were excluded as well. In addition, manually selected papers from references within selected researches were included in this paper. 

## 3. Results

After an electronic search, the total number of studies that investigated functional movement via functional movement screen was 202. After inclusion and exclusion criteria were met, and after manually selection of references, 14 studies were included in this narrative review. After search, studies were categorized into three distinct areas: (1) functional movement in the average adolescent population and exercise intervention aimed to improve functional movement outcomes (six studies selected), (2) physical activity, weight status and functional movement among the adolescent population (four studies) and (3) maturation and functional movement among the athletic population of adolescents (four studies). Each of these study areas are incorporated and interpreted in the discussion section.

## 4. Discussion

### 4.1. Clinical Importance of the Functional Movement and Functional Movement Screen as a Diagnostic Tool

The Functional Movement Screen^TM^ (FMS^TM^), originally created by Gray Cook and Lee Burton, is a screening instrument intended to evaluate deficiencies in mobility and stability [12,13]. FMS^TM^ includes seven tests: the deep squat, hurdle step, inline lunge, shoulder mobility, ASLR, trunk stability push-up and rotary stability. To the author’s knowledge, this is the only diagnostic instrument available, described and validated in scientific literature for the purpose of screening functional movement patterns. 

Scoring the FMS^TM^ has its own rules and procedures. The FMS raters use a standardized procedure to evaluate movement function. While performing FMS testing, each participant has a maximum of three trials for each test in accordance with the recommended protocol. Each test is scored on a three-point scale, from 0 to 3, with higher scores indicating better FM. In the presence of pain, a score of zero is noted. For each test, the highest score from three trials is recorded. An overall composite score was calculated with a total FMS score of 21 according to standardized guidelines. Descriptions of each FMS test, as well as standardized guidelines for the complete FMS testing is well written in the literature and can be studied elsewhere [12,13].

Some studies have shown the efficiency of the FMS in determining injury risk in athletes [14,15,16], however, others indicated the opposite [17,18,19]. Although there is conflicting evidence about the FMS as an injury predictive tool, the author’s opinion is that the FMS has critical value for identifying movement mobility and stability deficiencies. Deficits in movement mobility and joint stability potentially predispose athletes and average populations to higher injury risk since optimal movement patterns can possibly prevent and reduce that risk. 

Several studies reported moderate to good inter-rater and intra-rater reliability of the FMS even among novice raters [20,21]. In addition, two-hour education on using FMS as a diagnostic tool seems to be efficient according to prior research [21]. 

### 4.2. Functional Movement in the Average Adolescent Population and Exercise Intervention aimed to improve Functional Movement Outcomes

Although there are a number of studies that investigated functional movement among athletic adolescents, only few studies investigated FM in the average adolescent population. These studies investigated relations between FM and PA or weight status. Additionally, few studies investigated maturation effect on functional movement in the athletic adolescent population. Only one study provided normative values for the FMS in the adolescent population [2]. Another study investigated the prevalence of functional movement patterns in children over the first three years of post-primary education [3].

Abraham et al. [2] investigated functional movement patterns in the average population of adolescents. This study included a large number of participants (*n* = 1005) with ages from 10 to 17 years old. They reported a mean value of the total FMS score of 14.5 points. Additionally, results showed a significant difference in total FMS score between females and males. However, no significant difference in scores existed between those who reported a previous injury and those who did not. In addition, the authors suggested normative values for the individual functional movement patterns for this population that can be found in their paper [2]. However, there are few limitations of this study that should be considered while implementing normative values in the practice. Large age span (10–17 years) among participants in this study reveals that pre-pubertal and pubertal subjects were included in the sample. Additionally, this study excluded all inactive children which could potentially lead to higher mean values. Although there are few limitations in this study, this is the first study that provided normative values for a school aged adolescent population.

Lester et al. [3] examined the age-related association of functional movement among children in Ireland. The aim of this study was to gather data on prevalence of movement skills and functional movement patterns in children over the first three years of post-primary education (*n* = 181, mean age = 14.4). In this research, 43.6% of adolescents were in year one, 23.8% of adolescents were in year two and 32.6% adolescents were in year three with age range from 12.3 to 16.4 years old. Looking altogether, authors reported that results of the functional movement outcomes in their sample were suboptimal across all years. As authors stated, significant age-related differences were reported. When we look this data, as age increases, scores on the in-line lunge pattern decrease (difference between the first and third year). Additionally, the mean total FMS score reported in this study was 14.05, which is similar to results obtained by Abraham et al. The authors of this study strongly suggest that school-based intervention should be incorporated across the post-primary education child population in order to decrease decline in the impaired movement patterns.

Four studies investigated the impact of exercise intervention on functional movement outcomes. Coker [4] investigated the impact of the standardized warm-up protocol in middle school children on functional movement parameters. Participants from seventh-grade and four eighth-grade physical education classes participated in this study (*n* = 120, mean age = 13.1 years old). A six-week intervention included exercises that targeted mobility and stability of joints and muscle activation (exercise targeted ankle joint mobility, pelvic stability and dysfunctional gluteal, abductors, and adductors muscles). Results of this study suggest that a warm-up, which consists of exercises that target typical movement and body dysfunctions among adolescents of a sensitive age, can significantly reduce dysfunctional movement patterns. It is the author’s opinion that school policies should implement these programs into the physical education curriculum in order to reduce dysfunctional movement pattern prevalence and potentially reduce risk injury incidence among the average adolescent population.

A study done by Nourse et al. [5] investigated the impact of live video diet and exercise intervention on vascular and functional outcomes in overweight and obese children (mean age = 14.5, *n* = 20). The intervention lasted 12 weeks and included three times per week videoconferences with a trainer and diet consultations. Results of this study showed a significant reduction in waist-hip ratio and improvement in total functional movement screen score. Average improvement of the participants in total FMS score was 13 to 17 points, from baseline to the end of this intervention, respectively. Authors of study concluded that a 12-week live video intervention improves functional movement outcomes in the population of overweight and obese adolescents.

St Laurent et al. [6] investigated the impact of a suspension-training movement on functional movement in children (*n* = 28, average age = 9.3 years old). Participants were divided into two groups (control and intervention group). After the six-week suspension-training movement program was finished, the intervention group showed better results in functional movement outcomes relative to the control group. Authors of this study suggest that intervention using this kind of training modality could be beneficial for improving functional movement outcomes.

Wright et al. [7] examined the impact of fundamental movement training on functional movement outcomes in physically active children (*n* = 22, average age = 13.4). Participants were divided into two groups, where the intervention group was included in the training that focused on movement quality (weekly 4 × 30-min session) and participants from the control group were involved in multisport activity. Interestingly, results showed that short-term intervention focusing on movement quality did not have an effect on functional movement parameters in physically active children compared to the control group.

### 4.3. Physical Activity, Weight Status, and Functional Movement among Adolescent Populations

Although PA level and FM have critically important roles for general health of children, to date, only three studies appear to have examined relations between FM, weight status, and PA level among adolescents [8,9,10,11].

Study performed in Moldova investigated the relationship between FM, core strength, posture, and body mass index (BMI) [8]. Researchers collected data from 77 children, from 8 to 11 years old with an average BMI value of 16.4. Mitchell et al. reported the average total FMS score of 14.9 [8], which is slightly higher than the study done by Abraham et al. [2]. The results of this study showed that static posture and BMI are not related to FM. Additionally, researchers did not find a correlation between posture and FMS total score. These results are obvious since posture in this study was assessed in the static position while FMS tests assess movement and dynamic postural stabilization. On the other hand, results showed that core strength was positively related to the total FMS score. The authors of the study concluded that the individual test scores indicate that none of the test items were too difficult for the children, which means that the same tests can be directly used in clinical practice and school classes with school-aged children.

An interesting study done by Duncan et al. [9] examined the association between FM and overweight and obesity in British children. Data were obtained from 90 children, 7–10 years old. After BMI was determined, children were classified as normal weight, overweight, or obese according to international official guidelines. The results for total FMS score for normal weight children was 14.7, for overweight 12.2, and for obese children 9.0. Duncan et al., showed that total FMS score was negatively correlated with BMI. Additionally, the scores in all individual FMS tests were higher for normal weight children compared to obese children. In addition, normal weight children performed better than overweight children in the two tests: deep squat and shoulder mobility. On the other hand, overweight children scored better than obese children in four movement patterns (hurdle step, inline lunge, shoulder mobility, and ASLR). This result puts overweight and obese children in the group of children with increased risk for injury incidence. These are clinically important findings, because over time, dysfunctional movement patterns along with the effect of excess weight and consequently higher load on the joints, can possibly lead to degenerative changes in later life. This research highlights that overweight and obesity are significantly associated with poorer functional movement in children. 

Findings of this study seems to be contradictory with the results of the study mentioned before [8]. However, in a study done by Mitchell et al., there are few limitations that need to be considered. The authors did not separate participants into three categories (normal weight, overweight, and obese). In addition, 9% of the children were categorized as overweight (with no information about number and percentage of obese children), whereas in study done by Duncan et al. one third of children were classified as overweight/obese [9]. This limitation can potentially lead to opposite results and limited conclusion, and therefore can minimize the importance of potential relations between higher values of the BMI and suboptimal functional movement patterns in children.

In this research field, one more interesting study was performed by the same authors [10]. Duncan and Stanley investigated relations between weight status, physical activity level, and functional movement in British children. This study was performed on 58, 10–11 year old children. The results showed that the total FMS score was negatively correlated with BMI and positively related to PA level. Normal weight children scored significantly better for total FMS score compared to children classified as overweight/obese. The mean of total FMS scores was 15.5 for normal weight children and 10.6 in overweight/obese children. 

Duncan and Stanley explained these results through few possible mechanisms. These authors suggested that deficits in FM could exist prior to being overweight. They pointed out that: “Excess weight and functional prowess are the results of natural selection since children who are functionally limited will remain inactive and will not develop optimal functional movement patterns that underpin performance to the same level of mastery as children without functional limitation.” [10]. Additionally, the authors discussed that children who are not functionally limited may more likely enjoy PA, and thus, engage in more regular practice of functional movement patterns that underpin performance. Looking altogether, the results presented in these studies support the need for interventions to increase level of physical activity and improve functional movement in overweight and obese pediatric populations.

Garcia-Pinillos et al. [11] examined relations between functional movement patterns and weight status in children aged between 6 and 13 years old (*n* = 333). Results of this study show that weight status is moderately negatively correlated with total FMS score. In addition, overweight and obese children showed poorer functional movement compared to normal weight children. These results are consistent with the study done by Duncan et al. In addition, significant differences were found between normal weight, overweight, and obese children in lower-extremity movement patterns (deep squat, hurdle step, in-line lunge) and flexibility tests (shoulder mobility, straight leg-raise), but also in trunk stability pattern (push-up). This research revealed that girls outperformed boys in tests that require flexibility and balance, while boys outperformed girls in stability tests which support previous findings in context of sex dimorphism in individual functional movement patterns [2].

The information presented in the paragraphs above are essentially important for the practice of physical education teachers, coaches, and other professionals who work with pediatric populations. Optimal level of PA and optimal FM in children can reduce the risks of orthopedic abnormality arising from suboptimal movement patterns in later life. Since suboptimal movement patterns and low PA level could predispose children to a higher risk of the injury incidence, practitioners should consider functional movement interventions. It is the author’s opinion that this population needs specific exercises that address suboptimal movement patterns first, and then exercises targeting weight status to minimize risk of high-load exercise on the skeletal system in the pediatric population.

### 4.4. Maturation and Functional Movement among Athletic Populations of Adolescents

To date, a number of studies investigated maturation effects in the average and athletic population of adolescents. However, only few studies appear to have examined the relationship between maturity and functional movement patterns [22,23,24,25]. These studies included only the athletic adolescent population. Within this paragraph, the author will briefly provide a review of current evidence and conclusions about maturation effects on functional movement among athletic adolescents.

A study done by Portas et al. investigated the effect of maturity on functional movement screen scores in elite, adolescent soccer players [22]. The authors showed that maturity has substantial effects on FMS performance. Although this research highlights that findings are relevant only to those analyzing movement of soccer players, the authors of the mentioned study concluded that FMS assessment appears to be invalid for practical usage for very young players. 

Paszkewicz et al. compared functional and static evaluation tools among adolescent athletes [23]. The authors of this study compared FMS scores and Beighton and Horan joint mobility index (BHJMI) scores among pubescence in adolescent athletes. Based on the results of the modified pubertal maturation observational scale, the authors separated subjects into three groups: prepubescent, early-pubescent, and postpubescent groups. The researchers revealed a main effect for FMS scores across pubertal groups, but not in BHJMI composite scores. The postpubescent participants had higher FMS scores compared with the prepubescent participants and the early-pubescent athletes. Additionally, the results of this study did not confirm any correlation between FMS composite scores and BHJMI composite scores. The results of this study suggest that the FMS can discriminate between levels of pubescence and detect alterations during the pubertal growth cycle, whereas the BHJMI may not. 

Lloyd et al. examined relationships between functional movement screen scores, maturation, and physical performance in young soccer players [24]. This study demonstrated that variation of physical performance of youth soccer players could be explained by a combination of both functional movement screen scores and maturation. 

Wright and Chesterton [25] aimed to investigate differences between individual functional movement patterns at different stages of maturation in young athletes (mean age = 14.1 years, age between 8 and 18 years old) from various sports (field athletics, endurance sport, team sports, combat, and water sports). Participants were categorized in the three distinct maturation groups, participants who were before, at, and after their adolescent growth spurt (peak height velocity (PHV)). The authors found that differences among these groups were greatest in movement patterns that have high demands on stability, which suggests that adolescents potentially develop stability in this period of growth. These findings are consistent with studies mentioned before in this section [22,23]. In addition, results of the push-up test were higher in children who were at growth spurt or after growth spurt, when compared with children before growth spurt. In addition, authors of this study concluded that maturation has no effect on total FMS score.

However, it appears that there is no study that investigated the relationship between maturation effects and functional movement patterns among the average adolescent population. Due to the importance of this research field, the author is of the opinion that more studies are necessary in this research field.

## 5. Conclusions

This paper gave a detailed description and summarization of the current literature in the field of pediatric PA level, obesity, and maturation related to functional movement. Although there are only few studies in this field of research, the author highlights the importance and health benefits of optimal FM in children, as well as the consequences of dysfunctional movement patterns on the health of the locomotor system. Summary of current evidence suggests that decreased physical activity level is negatively correlated to functional movement in adolescence. Additionally, most studies suggest that weight status is negatively correlated to functional movement patterns, although there is conflicting evidence in this area. Evidence consistently showed that overweight and obese adolescent exhibit poorer functional movement compared to normal weight adolescents. Most of the studies that examined effects of exercise intervention on functional movement improved functional movement outcomes, while one study showed the opposite. It is clear that more research is needed on this topic to establish true intervention effects. In addition, it appears that maturation has effects on functional movement in the athletic population of adolescents. It is therefore important that practitioners consider interventions which develop optimal functional movement alongside physical activity and weight management strategies in children, in order to reduce the risks of injuries and pathological abnormality arising from suboptimal movement patterns in later life.

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
