# Peer review of "Relation between Weight Status, Physical activity, Maturation, and Functional Movement in Adolescence: An Overview"

_jfmk, 2019, doi:10.3390/jfmk4020031_

Round 1

Reviewer 1 Report

The references sources are very few...please include more sources on this topic.

Also, please present other recent studies on this topic because in the article very few results are presented.

Arrange the article in accordance with the template of the journal. 

Author Response

Response to Reviewer 1 Comments

Dear reviewer,

Thank you for provided comments and suggestion. Your suggestions were helpful and greatly improved our paper. Our responses to your suggestions are written below and incorporated into paper.

Point 1: The references sources are very few...please include more sources on this topic.

Response 1: After searching strategies and inclusion/exclusion criteria, authors added more references in this article.

Point 2: Also, please present other recent studies on this topic because in the article very few results are presented.

Response 2: Authors incorporated seven recent studies after inclusion/exclusion criteria were met and incorporated them into this article.

Point 3: Arrange the article in accordance with the template of the journal. 

Response 3:  Sections ‘Materials and methods’ and ‘Results’ are now added in this review and article is now arranged according to the template of the journal.

Reviewer 2 Report

1.       Searching strategies for all the relevant studies for the over view were not highlighted.

2.       Inclusion and exclusion criteria of the studies could be helpful to some readers.

3.       Line 78-94- The authors review one study providing the rational behind such a review as compared to other studies. This could introduce bias in the interpretation and the conclusion drawn.

4.       Page 2 line 55- The author describe the screening tool for functional movements. It is not clear if this is the only test used in the literature for functional movements or the rational for selection of this screening test will interest some readers.

5.       Page 4- line 127-134- The authors did not provide the overall relationship of functional movements and kinesiology. Only a few selected studies were selected without their  justification

6.       Explanation of the major findings of the study  were not adequately addressed.

7.       Selection of studies on adolescent, young soccer players as oppose to other professional athletes in other sporting code needs clarification.

Author Response

Response to Reviewer 2 Comments

Dear reviewer,

Thank you for the provided comments and suggestion. Your suggestions were helpful and greatly improved our paper. Our responses to your suggestions are written below and incorporated into the paper.

Point 1: Searching strategies for all the relevant studies for the overview were not highlighted.

Response 1: Searching strategies are now added in ‘’Materials and methods’’ section.

Point 2: Inclusion and exclusion criteria of the studies could be helpful to some readers.

Response 2:  Inclusion and exclusion criteria are now added in ‘’Materials and methods’’ section.

Point 3:  Line 78-94- The authors review one study providing the rational behind such a review as compared to other studies. This could introduce bias in the interpretation and the conclusion drawn.

Response 3: After searching and inclusion/exclusion criteria were added into this paper, authors incorporated five more studies that met criteria and incorporated into section 4.2.: ‘’Functional Movement in Average Adolescent Population and Exercise Intervention aimed to improve Functional Movement Outcomes’’.

Point 4: Page 2 line 55- The author describe the screening tool for functional movements. It is not clear if this is the only test used in the literature for functional movements or the rational for selection of this screening test will interest some readers.

Response 4: To the author's knowledge, this is an only diagnostic instrument available, described and validated in the scientific literature for the purpose of the screening functional movement patterns.

Point 5: Page 4- line 127-134- The authors did not provide the overall relationship of functional movements and kinesiology. Only a few selected studies were selected without their  justification.

Response 5: After searching and inclusion/exclusion criteria were added into this paper, the authors incorporated one more research that met criteria and incorporated into this section.

Point 6:  Explanation of the major findings of the study  were not adequately addressed.

Response 6: Authors added a more detailed explanation of the major findings into ''Abstract'' and ''Conclusion'' section.

Point 7: Selection of studies on adolescent, young soccer players as oppose to other professional athletes in other sporting code needs clarification.

Response 7: After detailed searching, authors found one more study that investigated athletes in various sports. This study is now incorporated into the last section of the discussion.

Round 2

Reviewer 1 Report

-

Reviewer 2 Report

None